# WSEL: EEG feature selection with weighted self-expression learning for incomplete multi-dimensional emotion recognition

## ABSTRACT

Due to the small size of valid samples, multi-source EEG features with high dimensionality can easily cause problems such as overfitting and poor real-time performance of the emotion recognition classifier. Feature selection has been demonstrated as an effective means to solve these problems. Current EEG feature selection research assumes that all dimensions of emotional labels are complete. However, owing to the open acquisition environment, subjective variability, and border ambiguity of individual perceptions of emotion, the training data in the practical application often includes missing information, i.e., multi-dimensional emotional labels of several instances are incomplete. The aforementioned incomplete information directly restricts the accurate construction of the EEG feature selection model for multi-dimensional emotion recognition. To wrestle with the aforementioned problem, we propose a novel EEG feature selection model with weighted self-expression learning (WSEL). The model utilizes self-representation learning and least squares regression to reconstruct the label space through the second-order correlation and higher-order correlation within the multi-dimensional emotional labels and simultaneously realize the EEG feature subset selection under the incomplete information. We have utilized two multimedia-induced emotion datasets with EEG recordings, DREAMER and DEAP, to confirm the effectiveness of WSEL in the partial multi-dimensional emotional feature selection challenge. Compared to nine state-of-the-art feature selection approaches, the experimental results demonstrate that the EEG feature subsets chosen by WSEL can achieve optimal performance in terms of six performance metrics.

## CCS CONCEPTS

• **Computing methodologies** → **Cognitive science**; **Feature selection**; • **Human-centered computing** → **HCI design and evaluation methods**.

## KEYWORDS

Incomplete multi-dimensional emotional labels, affective computing, feature selection, EEG, weighted self-expression learning

## 1 INTRODUCTION

Electroencephalogram (EEG) is a portable, non-traumatic technology for measuring brain activity that can react quickly to a range of affective states[32, 42]. EEG-based emotion recognition has received a lot of interest lately in the field of multimedia-induced affective computing because of its high temporal resolution and affordability[6, 9, 23]. Various types of feature extraction approaches, such as non-stationary index (NSI)[19], rational asymmetry (RASM)[26], higher-order crossing (HOC)[33], etc., have been used to analyze the non-stationary and nonlinear EEG signals in order to correctly represent different affective states.

The quantity of electrodes accessible for emotion recognition is rapidly increasing due to the development of EEG signal collecting equipment, and a large number of EEG features could be extracted from the electrodes [3, 40]. However, owing to the relatively limited number of EEG samples, the related features are often high-dimensional and always consist of redundant, irrelevant, and noisy information, which may significantly decrease the emotion recognition performance [39]. Feature selection is an effective means to choose noteworthy features and exclude irrelevant information from the original features, which retains the original neural representation information of the EEG features and enhances the transparency and interpretability of the emotion recognition model[13].

The EEG feature selection algorithms could be broadly divided into three types based on feature subset assessment and search mechanism: filter, wrapper, and embedded approaches [12]. The filter techniques consider the statistical characteristics of the EEG data to evaluate the significance of EEG features in affective computing. However, the feature selection performance is often unsatisfactory with these techniques, regardless of the learning algorithm [53]. Several studies have attempted to utilize wrapper techniques to address the issue. In many cases, the wrapper techniques could achieve better prediction performance than the filter techniques[12] due to the fact they employ the learning outcomes of a particular classifier as the evaluation index for the EEG feature subset. Nevertheless, the wrapper approaches often require lots of trials and incur significant computing costs[53]. Recently, researchers have been paying close attention to embedded approaches as a potential substitute approach to deal with the filter problem. The search for EEG feature subsets can be integrated into the optimization

problem using embedded techniques. The effectiveness of the embedded techniques on the EEG-based emotion recognition task has been demonstrated [44, 45].

Current EEG feature selection research relies on the completeness of the data. However, the training data in the practical application often includes missing information, such as partial dimensions of emotion labels from several samples, owing to the open acquisition environment, subjective variability, and border ambiguity of individual perceptions of emotion. The above incomplete information directly restricts accurate modelling for the relationship between the EEG and the multi-dimensional emotional labels.

To address the issue, we propose a novel EEG feature selection model with weighted self-representation learning (WSEL) for incomplete multi-dimensional emotion recognition. The model could reconstruct the label space through the second-order correlation and higher-order correlation within the multi-dimensional labels and simultaneously select the informative and redundant EEG feature subsets under the partial multi-dimensional information.

Furthermore, the following are the primary contributions to our work:

- This study proposes an incomplete multi-dimensional emotion feature selection method. The method embeds the process of EEG feature selection into an extended weighted self-representation model, which is able to suppress the effect of missing labeling data on model construction and introduce second-order and higher-order correlations within multi-dimensional emotion labels to recover missing labels.
- In order to copy the optimization problem of WSEL, an efficient and straightforward alternative scheme is proposed to ensure convergence and obtain an optimal solution.
- To confirm the effectiveness of WSEL in the incomplete multi-dimensional emotion feature selection challenge, two multimedia-induced emotion datasets with EEG recordings and multi-dimensional emotional labels, DREAMER and DEAP, have been implemented. Compared to nine state-of-the-art feature selection approaches, the experimental outcomes demonstrate that the EEG feature subsets chosen by WSEL can achieve the best emotion recognition performance on six evaluation metrics.

## 2 NOTATIONS AND RELATED WORKS

### 2.1 Notations and definitions

This section provides a concise summary of the definitions of the norms and symbols adopted throughout the work. Vectors are represented by lowercase strong letters ($\mathbf{x}$, $\mathbf{y}$ ...), whereas matrices are represented by capital letters ($X$, $Y$ ...). The mathematical notation for the transpose is represented by an uppercase superscript $T$. The operator denoted by $\odot$ represents the Hadamard product. The trace of a matrix is represented as Tr. The Frobenius norm and $l_{2,1}$ norm of a matrix $X$ are represented as:

$$\|X\|_F = \sqrt{\sum_{i=1}^{d}\sum_{j=1}^{n} x_{ij}^2} = \sqrt{tr\left(X^T X\right)} \tag{1}$$

$$\|X\|_{2,1} = \sum_{i=1}^{d} \sqrt{\sum_{j=1}^{n} x_{i,j}^2} = \sum_{i=1}^{d} \|x_{i,:}\|_2 \tag{2}$$

$X \in \mathbb{R}^{d \times n}$ is an EEG feature matrix, and each row represents a feature vector $\mathbf{x}_d \in \mathbb{R}^{1 \times n}$. $Y \in \{-1, 1\}^{n \times k}$ is a multi-dimensional emotional label matrix. The variables $d$, $n$, and $k$ correspond to the quantities of features, samples, and dimensions, respectively. The vector $\mathbf{1}_n = (1, 1 \ldots, 1)^T \in \mathbb{R}^{n \times 1}$ is defined as a column vector consisting of all elements being equal to one. The symbol $I_n \in \mathbb{R}^{n \times n}$ represents an identity matrix.

### 2.2 Emotion representation models

Emotion representation models fall into two categories: discrete emotion models and dimensional emotion models, depending on how emotions are depicted. Discrete emotion models utilize several basic discrete categories to characterize emotions. For example, there are six basic discrete categories, namely happiness, sadness, fear, anger, disgust, and surprise [34]. Complex emotion categories are composed of combinations of the basic discrete categories. However, the representation approach cannot scientifically describe the nature of emotions or effectively quantify emotional states from a computational perspective [1].

In order to solve the difficulties faced by the discrete model, the dimensional affective model maps affective states as points on a multi-dimensional space, and the affective states are distributed in different locations in the space based on different dimensions, and the distances between the locations reflect the differences and associations between different affective states [1]. Compared with the discrete emotion model, the dimensional emotion model has the advantages of a wide range of characterization and the ability to describe the emotional evolution process [8]. Commonly used dimensional models include the Valence-Arousal (VA) model [35] and the Valence-Arousal-Dominance (VAD) model [29].

### 2.3 EEG feature selection methods

The EEG feature selection techniques may be categorized into three groups based on their interaction with classification models: filter-based, wrapper-based, and embedded-based methods [53].

Filter-based methods assess the significance of EEG features based on specific criteria. The discriminatory EEG features that score highly are subsequently chosen, such as ReliefF [49], information gain [4], maximum relevance minimum redundancy [2, 41], conditional mutual information maximization [47], etc. However, the aforementioned filter-based methods may overlook a number of helpful features

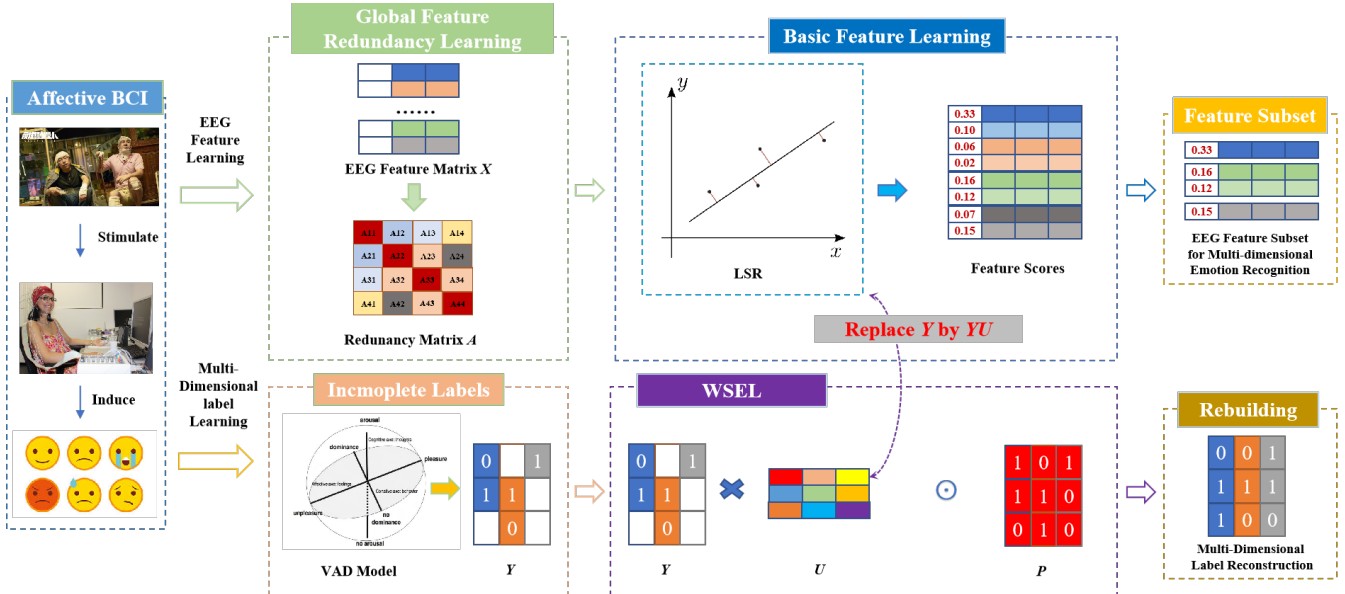

**Figure 1: The proposed WSEL framework concludes the following three sections: (a) basic feature learning; (b) weighted self-expression learning (WSEL); and (c) global feature redundancy learning.**

that, when paired with other features, are discriminative because learning models are neglected [36].

Using random or sequential search, wrapper-based approaches "wrap" features into candidate EEG feature subsets. A learning or prediction model then assesses the performance of these candidate EEG feature subsets. Several popular wrapper-based techniques, including the evolutionary computation algorithm [30] and the ReliefF-based genetic algorithm [18], have been proposed or employed for EEG-based affective computing. Nevertheless, the computational complexity of wrapper-based techniques is often significant because of the iterative nature of feature subset searches [37].

In order to address the shortcomings of filter-based or wrapper-based feature selection techniques, embedded-based methods have been put forward [24]. These methods integrate feature selection into the training model process and assess the relative significance of individual EEG features while optimizing learning models. Because of its completeness in statistics theory and efficacy for data analysis, least squares regression is a popular statistical analysis model for embedded-based feature selection approaches [5, 31, 43, 44]. Least square regression (LSR)-based embedded feature selection algorithms seek to learn a projection matrix $W$ and then rank the feature significance by $\{\|w^1\|_2, ..., \|w^d\|_2\}$ [46]. The discriminative EEG feature subsets can be chosen according to their significance.

## 3  PROBLEM FORMULATION

This section proposes a unique EEG emotional feature selection approach to acquire non-redundant and relevant feature subsets for incomplete multi-dimensional affective computing.

The WSEL framework is defined as follows:

$$\min_{W,U} F(X, W, Y, U) + \lambda C(Y, U) + \gamma \Omega(W) \quad (3)$$

where the projection matrix and coefficient matrix are represented by the variables $W$ and $U$, respectively. $\gamma$ and $\lambda$ are both tradeoff parameters. The basic feature learning function, weighted self-expression learning function, and global feature redundancy learning function are denoted by the symbols $F$, $C$, and $\Omega$, respectively. The next sections will introduce the definitions of $F$, $C$, and $\Omega$.

### 3.1  Basic feature learning

LSR is employed to estimate the correlation between the EEG feature matrix $X$ and the self-expression matrix $YU$. To perform EEG feature selection, the projection matrix $W$ is constrained by $l_{2,1}$-norm. The term $F(X, W, Y, U)$ could be expressed as follows:

$$F(X, W, Y, U) = \left\| X^T W + \mathbf{1}_n \boldsymbol{b}^T - YU \right\|_F^2 + \delta \|W\|_{2,1} \quad (4)$$

in which $\boldsymbol{b} \in \mathbb{R}^{k \times 1}$ is a bias vector, $W \in \mathbb{R}^{d \times k}$ is the projection matrix, $U \in \mathbb{R}^{k \times k}$ is the coefficient matrix, and $\delta$ ($\delta > 0$) is a tradeoff parameter.

### 3.2  Weighted self-expression learning

To describe the dimension-missing information, the element $P_{i,j}$ of an indicator matrix $P$ is defined as follows:

$$P_{ij} = \begin{cases} 1 & \text{if } i\text{-th instance exists in } j\text{-th dimensional label;} \\ 0 & \text{otherwise.} \end{cases}$$

$$(5)$$

Then, $P$ is introduced into the self-expression learning function as follows:

$$\min_U \|P \odot (Y - YU)\|_F^2 + \beta\|U\|_{2,1} \tag{6}$$

Via high-order label correlations among the multi-dimensional emotion labels, each dimension can be represented by all the dimensions. In order that each label is reconstructed by the most directly relevant labels, the self-expression coefficient matrix $U$ is imposed with a $l_{2,1}$-norm constraint.

Afterwards, a graph-based manifold regularizer is implemented to maintain consistency of the local geometry structures between the self-expression space $YU$ and the original multi-dimensional label space $Y$ [14]. A graph Laplacian matrix $L_Y \in \mathbb{R}^{n \times n}$ is indicated by the notation $L_Y = G - S$. The affinity graph for $Y$ is represented by $S$, and the elements of the diagonal matrix $G$ can be obtained by $G_{ii} = \sum_{j=1}^{n} S_{ij}$. A heat kernel has been employed to generate the affinity graph $S$. The similarity value of two labels, $\boldsymbol{y}_{i.}$ and $\boldsymbol{y}_{j.}$, is represented by the element $S_{ij}$. $S_{ij}$ is defined as follows:

$$S_{ij} = \begin{cases} \exp\left(-\frac{\|\boldsymbol{y}_{i.} - \boldsymbol{y}_{j.}\|^2}{\sigma^2}\right) & \boldsymbol{y}_{i.} \in \mathcal{N}_q\left(\boldsymbol{y}_{j.}\right) \text{ or } \boldsymbol{y}_{j.} \in \mathcal{N}_q\left(\boldsymbol{y}_{i.}\right) \\ 0 & \text{otherwise} \end{cases} \tag{7}$$

The symbol $\sigma$ represents the graph construction parameter, whereas $\mathcal{N}_p\left(\boldsymbol{y}_j\right)$ denotes the set of the top $q$ closest neighbours of the label $\boldsymbol{y}_{j.}$.

Ultimately, by including the graph-based manifold regularizer in the weighted self-expression learning, the formulation of the term $C$ can be expressed as follows:

$$\begin{aligned} C(Y, U) =& \|P \odot (Y - YU)\|_F^2 + \eta \operatorname{Tr}\left((YU)^T L_Y (YU)\right) \\ & + \beta\|U\|_{2,1} \\ & \text{s.t. } U \geq 0 \end{aligned} \tag{8}$$

where $\beta$ and $\eta$ represent tradeoff parameters.

### 3.3 Global feature redundancy learning

In addition, we introduce a global feature redundancy matrix $A$ to evaluate the correlations among all the EEG features. The value of $A$ could be obtained via the following calculation:

$$A_{i,j} = (O_{i,j})^2 = \left(\frac{\boldsymbol{f}_i^T \boldsymbol{f}_j}{\|\boldsymbol{f}_i\| \|\boldsymbol{f}_j\|}\right)^2 \tag{9}$$

where $\boldsymbol{f}_i \in \mathbb{R}^{n \times 1}$ and $\boldsymbol{f}_j \in \mathbb{R}^{n \times 1}$ are $i$-th and $j$-th centralized features of two types of EEG features $\boldsymbol{x}_i$ and $\boldsymbol{x}_j$ ($i, j = 1, 2, ..., d$). $\boldsymbol{f}_i$ and $\boldsymbol{f}_j$ can be computed as follows:

$$\begin{cases} \boldsymbol{f}_i = H\boldsymbol{x}_i^T \\ \boldsymbol{f}_j = H\boldsymbol{x}_j^T \end{cases} \tag{10}$$

where $H = I_{\mathrm{n}} - \frac{1}{n}\mathbf{1}_n\mathbf{1}_n^T$. Eq. (9) can be changed to

$$O = ZF^T FZ = (FZ)^T FZ \tag{11}$$

$F$ is defined as $F = [\boldsymbol{f}_1, \boldsymbol{f}_2, ..., \boldsymbol{f}_d]$. Let $Z$ be a diagonal matrix such that each diagonal element $Z_{i,i} = \frac{1}{\|\boldsymbol{f}_i\|}$ ($i = 1, 2, ..., d$). Since the matrix $O$ is positive semi-definite and $A = O \odot O$, the resulting global feature redundancy matrix $A$ is both non-negative and positive semi-definite [38]. Therefore,

the global feature redundancy learning function $\Omega$ can be defined as:

$$\Omega(W) = \operatorname{Tr}\left(W^T A W\right) \tag{12}$$

### 3.4 The final objective function of WSEL

By combining Eq. (4), Eq. (8), and Eq. (12) together, the proposed WSEL can be formulated as follows:

$$\begin{aligned} \min_{W,b,U} & \left\|X^T W + \mathbf{1}_n \boldsymbol{b}^T - YU\right\|_F^2 + \lambda\|P \odot (Y - YU)\|_F^2 + \beta\|U\|_{2,1} \\ & + \eta \operatorname{Tr}\left((YU)^T L_Y (YU)\right) + \mu \operatorname{Tr}\left(W^T A W\right) + \delta\|W\|_{2,1} \\ & \text{s.t. } U \geq 0 \end{aligned} \tag{13}$$

where $\lambda, \beta, \eta, \mu$, and $\delta$ are regularization parameters. The flowchart of WSEL is shown in Fig. 1.

## 4 OPTIMIZATION STRATEGY

Taking the partial derivative of Eq. (13) with respect to $\boldsymbol{b}$ and setting it equal to zero, we can solve for $\boldsymbol{b}$ using the equation $\boldsymbol{b} = \frac{1}{n}\left(U^T Y^T \mathbf{1}_n - W^T X \mathbf{1}_n\right)$. By replacing the aforementioned equation in Eq. (13), the optimization problem will be transformed to

$$\begin{aligned} \min_{W,U} & \left\|HX^T W - HYU\right\|_F^2 + \lambda\|P \odot (Y - YU)\|_F^2 + \beta\|U\|_{2,1} \\ & + \eta \operatorname{Tr}\left((YU)^T L_Y (YU)\right) + \mu \operatorname{Tr}\left(W^T A W\right) + \delta\|W\|_{2,1} \\ & \text{s.t. } U \geq 0 \end{aligned} \tag{14}$$

The alternatively iterative update technique is adopted to derive solutions for the two variables ($W$ and $U$) in Eq. (14). The technique is presented as follows:

### 4.1 Update $W$ by fixing $U$

When $U$ is fixed and we remove the irrelevant terms, we obtain the following function about $W$:

$$\mathcal{L}(W) = \left\|HX^T W - HYU\right\|_F^2 + \mu \operatorname{Tr}\left(W^T A W\right) + \delta\|W\|_{2,1} \tag{15}$$

By taking the partial derivative of $\mathcal{L}(W)$ w.r.t.$W$, we could get

$$\frac{\partial \mathcal{L}(W)}{\partial W} = 2XHX^T - 2XHYUW + 2\mu AW + 2\delta DW \tag{16}$$

where $D$ is a diagonal matrix and the element of $D$ is calculated by $D_{ii} = \frac{1}{2\sqrt{W_i^T W_i + \epsilon}}(\epsilon \to 0)$.

Hence, the optimal solution $W$ can be updated as follows:

$$W = (XHX^T + \mu A + \delta D)^{-1}(XHYU) \tag{17}$$

### 4.2 Update $U$ by fixing $W$

When $W$ is fixed, by introducing a Lagrange multiplier $\boldsymbol{\Psi}$ for $U \geq 0$, we have the following Lagrange function:

$$\begin{aligned} \mathcal{L}(U) = & \left\|HX^T W - HYU\right\|_F^2 + \lambda\|P \odot (Y - YU)\|_F^2 + \beta\|U\|_{2,1} \\ & + \eta \operatorname{Tr}\left((YU)^T L_Y (YU)\right) + \operatorname{Tr}\left(\boldsymbol{\Psi}^T U\right) \end{aligned} \tag{18}$$

Then, the partial derivative of $\mathcal{L}(U)$ w.r.t. $U$ in Eq. (18) is calculated as:

$$\frac{\partial \mathcal{L}(U)}{\partial U} = -2Y^T H X^T W + 2Y^T H Y U + 2\lambda Y^T (P \odot (YU - Y))$$
$$2\eta Y^T L_Y Y U + 2\beta V U + \mathbf{\Psi} \tag{19}$$

where $V$ is a diagonal matrix and the element of $V$ is calculated by $V_{ii} = \frac{1}{2\sqrt{U_i^T U_i + \epsilon}} (\epsilon \to 0)$.

Based on the Karush-Kuhn-Tucker complementary condition $\mathbf{\Psi}_{ij} U_{ij} = 0$, the update rule for $U$ is

$$U \leftarrow U \odot \frac{Y^T H X^T W + \lambda Y^T (P \odot Y)}{Y^T H Y U + \eta Y^T L_Y Y U + \lambda Y^T P \odot (YU) + \beta V U} \tag{20}$$

---

**Algorithm 1** EEG feature selection with weighted self-expression learning for incomplete multi-dimensional emotion recognition

---

**Input:** EEG feature matrix $X \in \mathbb{R}^{d \times n}$, incomplete multi-dimensional emotional label matrix $Y \in \mathbb{R}^{n \times k}$, and indicator matrix $P \in \mathbb{R}^{n \times k}$.

**Output:** Return ranked EEG features.

1: Initial $H = I_n - \frac{1}{n} \mathbf{1}_n \mathbf{1}_n^T$. Initial $W$ and $U$ randomly.
2: **repeat**
3:    Update $D$ via $D_{ii} = \frac{1}{2\sqrt{W_i^T W_i + \epsilon}}$;
4:    Update $V$ via $V_{ii} = \frac{1}{2\sqrt{U_i^T U_i + \epsilon}}$;
5:    Update $W$ via $W = (XHX^T + \mu A + \delta D)^{-1}(XHYU)$;
6:    Update $U$ via Eq. (20);
7: **until** Convergence;
8: **return** $W$ for EEG feature selection.
9: Sort the EEG features by $\|\boldsymbol{w}_i\|_2$;

---

Algorithm 1 provides the specific optimization steps for Eq. (13). The scores of each EEG feature in the incomplete multi-dimensional emotion recognition task can be evaluated by $W$. Ultimately, the optimal EEG feature subset with non-redundant and informative features is obtained.

## 5 EXPERIMENTAL DETAILS

This section introduces experimental details, including dataset description, EEG feature extraction, and experimental setup.

### 5.1 Dataset description

Comprehensive experiments are conducted on two EEG datasets with multi-dimensional labels to assess the performance of WSEL, including DEAP[17] and DREAMER[15]. The VAD paradigm is employed in the databases to describe human emotions. During the multimedia stimulation, EEG signals were concurrently recorded. A detailed description of the experimental setup can be found in [15] and [17]. Table 1 presents basic information for the two datasets.

The experiments used a band-pass filter with a cutoff frequency of 1-50 Hz to remove noise from the EEG recordings.

**Table 1: Comparisons between two EEG datasets.**

| Database | DREAMER | DEAP |
|---|---|---|
| Channel no. | 14 | 32 |
| Subject no. | 23 | 32 |
| Video no. | 18 | 40 |
| Sample no. | 414 | 1280 |
| Stimulus materials | film clips | music videos |

**Table 2: The dimensions of the thirteen EEG feature types extracted from the two databases.**

| Source domain | Feature type | DREAMER | DEAP |
|---|---|---|---|
|  | NSI | 14 | 32 |
|  | HOC | 140 | 320 |
| Time | SPE | 14 | 32 |
|  | SHE | 14 | 32 |
|  | C0 complexity | 14 | 32 |
|  | AP | 70 | 160 |
| Frequency | $AP_\beta / AP_\theta$ | 14 | 32 |
|  | DE | 70 | 160 |
| Time-frequency | AHTIMF | 70 | 160 |
|  | IPHTIMF | 70 | 160 |
|  | DASM | 35 | 70 |
| Spatial | RASM | 35 | 70 |
|  | FC | 91 | 496 |
| Total |  | 651 | 1756 |

Subsequently, artifacts related to eye movement and muscle activities were suppressed using independent component analysis. It should be mentioned that EEG feature extraction was performed on the whole trial as a sample. As stated differently, trials were not divided into many parts in order to expand the sample size of the experiments.

### 5.2 EEG feature extraction

Based on earlier studies on EEG emotional feature extraction and analysis [13, 45], thirteen types of EEG features were extracted for the multi-dimensional affective computing task, such as C0 complexity [54], NSI[19], differential asymmetry (DASM)[28], HOC[33], spectral entropy(SPE) [48], RASM [26], shannon entropy(SHE)[25], DE[7], absolute power ($AP$), the absolute power ratio of the theta band to the beta band ($AP_\beta / AP_\theta$) [16], the amplitude of the Hilbert transform of intrinsic mode functions (AHTIMF), the instantaneous phase of the Hilbert transform of intrinsic mode functions (IPHTIMF)[13], and function connectivity (FC). The comprehensive descriptions of the thirteen EEG features can be found in [7, 13, 44]. The dimensions of the thirteen EEG feature types are presented in Table 2.

## 5.3 Experimental setup

Nine advanced feature selection methods are compared in order to fully assess the performance of WSEL in the multi-dimensional affective computing task. The compared techniques include multi-label feature selection using multi-criteria decision making (MFS-MCDM) [10], multi-label graph-based feature selection (MGFS) [11], PMU[20], global relevance and redundancy optimization (GRRO) [50], scalable criterion for large label set(SCLS)[22], feature selection with orthogonal regression (FSOR) [45], global redundancy minimization in orthogonal regression (GRMOR)[44], FIMF[21], and feature selection method via manifold regularisation (MDFS)[51].

The EEG recordings were divided into low and high values depending on the self-assessed score of each affective dimension. The threshold value for this category was established at five. Multi-label k-nearest neighbor (ML-KNN) [52] was implemented as a base classifier. The values for the number of nearest neighbors and smooth were set to 10 and 1, respectively. Seventy percent of the participants were chosen randomly to be used as training sets, while the remaining thirty percent were designated as test sets. A cross-subject experiment setup was implemented. In order to mitigate any bias, a total of 50 separate and unbiased experiments were carried out, and the average outcome was regarded as the ultimate measure of affective computing.

The research applied six performance metrics for evaluating the effectiveness of multi-dimensional emotion recognition. These metrics include two label-based evaluation metrics, macro-F1 and micro-F1, as well as four example-based evaluation metrics: average precision, coverage, ranking loss, and hamming loss. The detailed information on the above metrics could be found in [51].

## 6 EXPERIMENTAL RESULTS AND DISCUSSION

### 6.1 Performance comparison in incomplete multi-dimensional emotion recognition

We employed the strategy in [27] to simulate the incomplete multi-label setting, eliminating certain ratios of labels from each dimension as missing data. With a step of 10% , the missing ratio was set between 10% and 50%. By using the aforementioned feature selection methods, almost 10% of all EEG features were selected. The tradeoff parameters ($\lambda$, $\beta$ $\eta$, $\mu$, and $\delta$) were tuned from $10^3$ to $10^3$ with a step of $10^1$.

Fig. 2 and Fig. 3 show the comparative results on the DEAP and DREAMER datasets. The horizontal axis of each figure represents the missing ratio of multi-dimensional labels, while the vertical axis represents the results of each performance metric. The results of WSEL are shown in Fig. 2 and Fig. 3 as the red line. As seen in Fig. 2(a-c) and Fig. 3(a-c), the multi-dimensional emotion recognition performance ability improves with decreasing values. Furthermore, as shown in Fig. 2(d-f) and Fig. 3(d-f), the multi-dimensional emotion

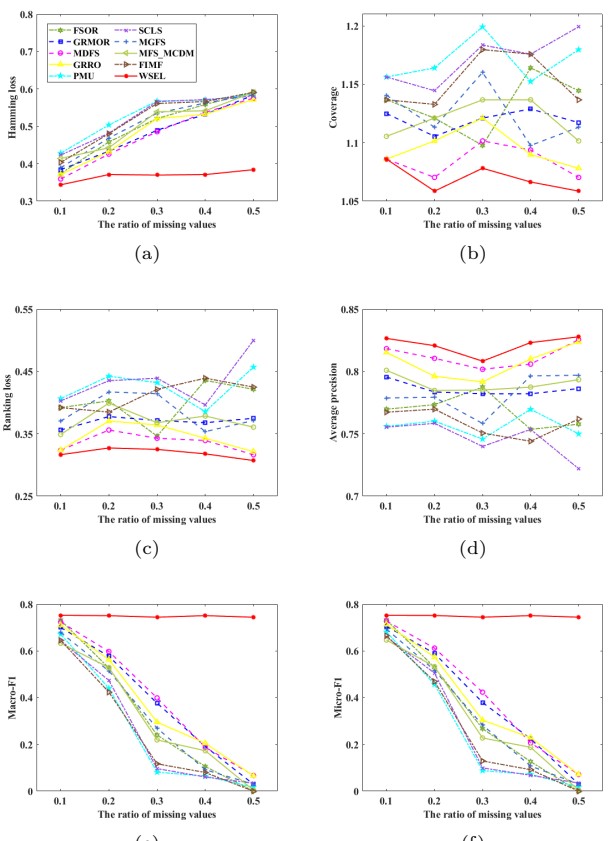

Figure 2: Multi-dimensional emotion classification performance with different missing ratios on the DEAP data set: (a) Hamming loss; (b) Coverage; (c) Ranking loss; (d) Average precision; (e) Macro-F1; (f) Micro-F1.

recognition performance ability improves with increasing values. Compared with the results of the other nine methods, WSEL almost maintains the best performance with various missing ratios.

Table 3 provides a summary of the quantitative comparative results across the six multi-label performance metrics. ↑ denotes a larger result being better, and ↓ denotes the contrary. Overall, the results in Fig. 2, Fig. 3, and Table 3 prove that the EEG feature subset chosen by the WSEL method has the best incomplete multi-dimensional emotion recognition performance.

To further validate the effectiveness of the weighted self-representation learning module for label reconstruction, the proportion of the number of successfully recovered emotional labels with various missing rates is summarized in Table 4. As shown in Table 4, the proposed method is able to effectively recover more than 60% of the missing labels through the weighted self-representation learning module, which in turn

**Table 3: The comparisons of average multi-dimensional emotion recognition results (↑ denotes a larger result being better, and ↓ denotes the contrary).**

| Methods | DEAP | | | | | | DREAMER | | | | | |
|---|---|---|---|---|---|---|---|---|---|---|---|---|
| | HL ↓ | RL ↓ | CV ↓ | AP↑ | MA↑ | MI ↑ | HL ↓ | RL ↓ | CV ↓ | AP ↑ | MA ↑ | MI ↑ |
| MDFS | 0.48 | 0.34 | 1.08 | 0.81 | 0.40 | 0.41 | 0.42 | 0.31 | 1.43 | 0.82 | 0.62 | 0.64 |
| GRRO | 0.49 | 0.34 | 1.10 | 0.81 | 0.37 | 0.38 | 0.41 | 0.34 | 1.45 | 0.81 | 0.62 | 0.64 |
| PMU | 0.53 | 0.42 | 1.17 | 0.76 | 0.25 | 0.26 | 0.48 | 0.42 | 1.53 | 0.77 | 0.48 | 0.51 |
| SCLS | 0.52 | 0.43 | 1.17 | 0.75 | 0.26 | 0.27 | 0.48 | 0.44 | 1.54 | 0.76 | 0.49 | 0.51 |
| MGFS | 0.51 | 0.39 | 1.13 | 0.78 | 0.32 | 0.33 | 0.47 | 0.43 | 1.53 | 0.77 | 0.53 | 0.56 |
| MFS_MCDM | 0.51 | 0.37 | 1.12 | 0.79 | 0.31 | 0.32 | 0.46 | 0.44 | 1.52 | 0.76 | 0.52 | 0.55 |
| FIMF | 0.52 | 0.41 | 1.15 | 0.76 | 0.25 | 0.27 | 0.49 | 0.47 | 1.56 | 0.75 | 0.47 | 0.51 |
| FSOR | 0.50 | 0.40 | 1.13 | 0.77 | 0.32 | 0.33 | 0.47 | 0.41 | 1.53 | 0.77 | 0.49 | 0.54 |
| GRMOR | 0.48 | 0.37 | 1.12 | 0.79 | 0.38 | 0.39 | 0.43 | 0.35 | 1.46 | 0.81 | 0.57 | 0.61 |
| **WSEL** | **0.37** | **0.32** | **1.07** | **0.83** | **0.75** | **0.75** | **0.28** | **0.28** | **1.39** | **0.84** | **0.83** | **0.83** |

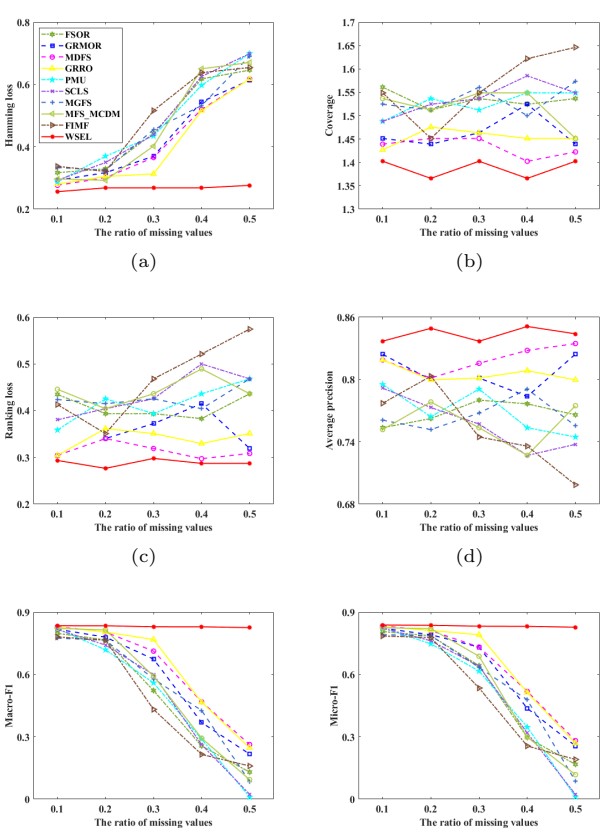

**Figure 3: Multi-dimensional emotion classification performance with different missing ratios on the DREAMER data set: (a) Hamming loss; (b) Coverage; (c) Ranking loss; (d) Average precision; (e) Macro-F1; (f) Micro-F1.**

provides richer multi-dimensional emotional label information for the basic feature learning module.

**Table 4: The accurate recovery rate(%) of missing labels.**

| Missing ratio | Accurate recovery rate | |
|---|---|---|
| | DEAP | DREAMER |
| 10% | 60.61 | 62.85 |
| 20% | 69.08 | 62.76 |
| 30% | 60.72 | 61.51 |
| 40% | 62.89 | 62.85 |
| 50% | 60.81 | 61.45 |

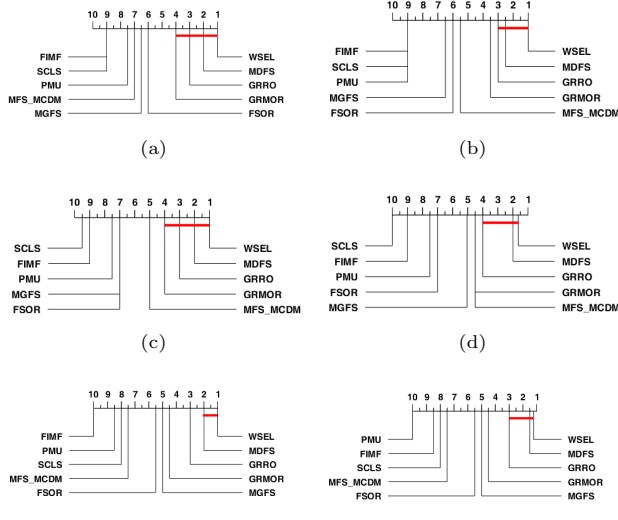

**Figure 4: The Nemenyi test results (significance level $\alpha = 0.05$): (a) Hamming loss; (b) Coverage; (c) Ranking loss; (d) Average precision; (e) Macro-F1; (f) Micro-F1.**

**Table 5: The results of ablation experiments on the performance index average precision (w/o, GMR, and GFRL denote without, graph-based manifold regularizer, and global feature redundancy learning, respectively).**

| Conditions | Average precision | |
|---|---|---|
| | DEAP | DREAMER |
| w/o WSEL | 0.75 | 0.78 |
| w/o GMR | 0.80 | 0.82 |
| w/o GFRL | 0.79 | 0.80 |
| Our method | 0.83 | 0.84 |

## 6.2 Nemenyi test and ablation experiments

The Nemenyi test is applied as a specific post-hoc test to complete the multi-dimensional emotion recognition performance comparison, and the WSEL is implemented as the control method. The outcomes of the Nemenyi test in terms of six performance metrics are shown in Fig. 4. WSEL outperforms the other nine approaches on all performance metrics, as Fig. 4 illustrates. This indicates that the WSEL method could achieve highly competitive multi-dimensional emotion recognition performance with incomplete label information.

To determine the contributions of each module in the proposed EEG feature selection model, we carried out ablation experiments. There are three important modules in WSEL, and we only sequentially removed each module. Table 5 illustrates that the weighted self-expression model plays a key role in recovering missing labels by efficiently capturing high-order correlation information within incomplete multi-dimensional emotional labels. Other modules play roles in maintaining the local geometry structures and removing redundant information.

**Table 6: The comparison of average computational time results (seconds).**

| Methods | DEAP | DREAMER |
|---|---|---|
| MDFS | 4.08 | 1.45 |
| GRRO | 32.42 | 2.01 |
| PMU | 134.19 | 7.54 |
| SCLS | 27.51 | 1.73 |
| MGFS | 0.45 | 0.34 |
| MFS_MCDM | 0.65 | 0.18 |
| FIMF | 0.03 | 0.01 |
| FSOR | 369.05 | 46.42 |
| GRMOR | 146.22 | 27.47 |
| **WSEL** | **3.98** | **0.99** |

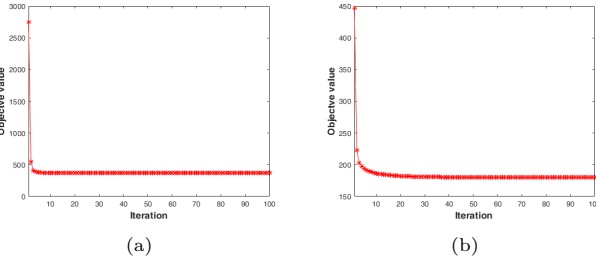

(a)                          (b)

**Figure 5: Convergence of the WSEL algorithm on the two data sets: (a) DEAP; (b) DREAMER.**

## 6.3 Computational cost and convergence demonstration

Additionally, the computational time cost of all the methods was compared. The implementation was made in MATLAB (MathWorks Inc., Novi, MI, USA) and run under Microsoft Windows $11 \times 64$ on a computer with an Intel Core i5-124000HQ 2.5 GHz CPU and 16.00 GB of RAM. The average computational time results are shown in Table 6. As seen in Table 6 and Table 4, the proposed method is able to obtain the optimal performance of incomplete multi-dimensional emotion recognition with relatively little computational cost.

Finally, we conduct a convergence speed analysis of the proposed iterative optimization algorithm. Fig. 5 shows the convergence curves of the objective value on the DEAP and DREAMER. The tradeoff parameters ($\lambda$, $\beta$ $\eta$, $\mu$, and $\delta$) are all fixed at 10. As seen in Fig. 5, the WSEL algorithm converges rapidly in a small number of iterations, demonstrating the potency of our iterative optimization strategy.

## 7 CONCLUSIONS AND FUTURE WORK

An embedded EEG feature selection framework under incomplete multi-dimensional emotion information is proposed, which selects discriminative EEG features and reconstructs the multi-dimensional emotion label space by simultaneously merging weighted self-expression learning and global feature redundancy learning in the least squares regression model. Furthermore, a simple and effective substitute strategy is also proposed to copy the optimization problem of WSEL. The experimental results have demonstrated the effective performance of WSEL.

Although the proposed method can recover missing emotional labels to a certain extent (more than 60%), its recovery rate is not particularly satisfactory, especially in real brain computer interface-based emotion recognition applications. It is an interesting topic to improve the recovery rate of missing labels. In the future, we will focus on researching novel strategies to improve the recovery rate of missing labels for incomplete multi-dimensional affective computing.

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
