# OpenReview forum: "WSEL: EEG feature selection with weighted self-expression learning for incomplete multi-dimensional emotion recognition"
_acmmm.org/ACMMM/2024/Conference — MM2024 Oral_

### Official Review · Reviewer_xq6x · 2024-05-17

**Rating:** 6
**Confidence:** 3

**Summary:**

This paper addresses the issue of incomplete labels that may arise in open scenes in EEG emotion recognition, enhancing feature selection methods. The approach includes three modules: basic feature learning, weighted self-expression learning, and global feature redundancy learning to accomplish the feature selection process. Specific optimization strategies are proposed, and the methods show state-of-the-art performance on two different multimodal emotion datasets.

**Strengths:**

1. Innovation: Compared to previous FSOR and GRMOR models, the WSEL module proposed in this paper is a significant innovation. It is used to describe and reconstruct multidimensional missing label information, providing a solution to the specific problem of missing labels.
2. Writing Clarity: The modules and descriptions in Figure 1 align well and are clearly articulated. Detailed descriptions are provided for different methods through formulas. Additionally, pseudocode is provided to explain the steps.
3. Review: The review of the three methods—filter, wrapper, and embedded—is thorough.
4. Significance: The explosion of feature volume is a significant issue in EEG emotion recognition. This paper further provides a solution to the problem of missing labels in open scenes, which is expected to promote the development of relevant sub-communities.
5. Experimental Validation: Analyses are conducted on various evaluation matrices, showing state-of-the-art performance.

**Limitations:**

1. Copyright Issue with Figures: The experimental scene diagram from the DEAP dataset homepage is pasted in Figure 1 (not affecting the model architecture). It is suggested that the authors consider potential copyright issues to avoid potential disputes.
2. Reproducibility: It is hoped that the authors will further open-source the code to promote development in related fields.

**Suitability:**

2

---

### Official Review · Reviewer_r3sa · 2024-05-20

**Rating:** 3
**Confidence:** 3

**Summary:**

The manuscript utilizes EEG data to implement emotion recognition.

**Strengths:**

Aiming at the problems of feature dimension catastrophe and incomplete multi-dimensional labeling information, the manuscript conducts multi-dimension emotion recognition research under incomplete labels and proposes a EEG feature selection method. Extensive experiments are conducted on two publicly available datasets to demonstrate the effectiveness of the method.

**Limitations:**

However, the innovation of the manuscript is somewhat weak and the experimental results are puzzling. The manuscript needs further improvement. The detailed comments are as follows:
1) From my point of view, the Figure 1 and Section 3 are confusing. As can be seen in Figure 1, EEG feature learning first implements global feature redundancy learning to obtain the EEG feature matrix X and redundancy matrix A. Then, the EEG feature subset is extracted by basic feature learning for multidimensional emotion recognition. However, in Section 3, basic feature learning is introduced first, then weighted self-expression learning, and finally global feature redundancy learning, which makes it difficult to understand the flow of the proposed model.
In addition, for Figure 1, the manuscript proposes a WSEL architecture, which consists of three sections, one of which is WSEL. Does WSEL represent the model framework or one of the methods?
2) Among the nine advanced feature selection methods, is there a method proposed for incomplete multi-dimensional labels?
3) Legends are missing in Figures 2 and 3. Figure 2 shows the emotion classification performance with different missing ratios. It is suggested to analyze the reason why the classification performance is almost unchanged in the case of larger missing ratios.
4) Table 2 does not indicate under which condition the emotion recognition results were obtained. It is recommended to compare with the following studies and describe their relevance and differences with the manuscript :
[1] Embedded EEG Feature Selection for Multi-Dimension Emotion Recognition via Local and Global Label Relevance,  IEEE Transactions on Neural Systems and Rehabilitation Engineering, vol. 32, pp. 514-526, 2024. (18 January 2024)
5) Table 4 exhibits the proportion of the number of successfully recovered emotional labels with various missing rates. Why when the missing rate is 40% and 50%, the accurate recovery rate of the label is higher than that of the missing rate is 10%?
6) The English should be improved, there are a number of problems with sentence grammar and structure and formatting errors, such as Eq. (13), (18) and (19).

**Suitability:**

2

---

### Official Review · Reviewer_sSZj · 2024-05-24

**Rating:** 4
**Confidence:** 3

**Summary:**

The paper introduces an EEG feature selection model called Weighted Self-Expression Learning (WSEL) for incomplete multi-dimensional emotion recognition. The primary goal of this model is to address the challenges posed by high-dimensional multi-source EEG features and the limited size of valid samples. WSEL performs feature selection and optimization by learning and utilizing the intrinsic structure and correlations within the data, which can suppress the influence of missing label data on model construction. The method has been tested on two well-known public datasets: DREAMER and DEAP.

**Strengths:**

- The proposed method, WSEL, embeds the EEG feature selection process into the extended weighted self-expression model, adopting High-Order Correlation to capture complex relationships among multiple variables, not limited to simple linear relationships.

- An effective and easy alternative is proposed to ensure convergence and obtain an optimal solution for Multi-Dimensional Emotion Recognition.

- Validated using two multimedia-induced affective datasets, DREAMER and DEAP, the WSEL-selected EEG feature subset performed best on all six performance indicators compared to nine existing feature selection methods.

**Limitations:**

#### **Major**:

- **Novelty**: The mathematical equations remind me of some current feature selection methods that were not included in the authors’ comparison study and other main contexts, such as IMUFS (Huang et al., IEEE TKDE, 2023) and the Joint Self-Expression Module (Yuan et al., PRCV 2020). Could the authors step-by-step explain their unique contributions compared to such methods to prove their novelty, both from a global conceptual perspective and with detailed mathematical explanations?

- The accurate recovery rates of missing labels vary irregularly with the missing ratio, rather than showing a negative correlation. Specifically, the highest recovery rate (69.08%) occurred at a 20% missing ratio. How did the authors consider their model’s unexpected degeneration when only 10% of the labels were missing?

- Since the motivation for this work, as stated at the beginning of the abstract, is to address overfitting, showing a comparison of training and testing loss under the conditions of no feature selection and with feature selection would be intuitive to indicate the power of the proposed methods.

- WSEL models involve multiple matrix operations and higher-order correlation calculations, which can be computationally expensive on large-scale datasets. How do the authors consider the computational efficiency of their method on large-scale datasets? The authors should interpret the reasons for the inferiority of the proposed method in terms of computational timing cost compared with models like MGFS, MFS, and FIMF, as well as discuss what causes the magnitude jump in the computational time of these models like PMU, FSOR, and GRMOR.

- The authors only show their own model’s convergence performance in Figure 5, which does not fully demonstrate the potency of the optimization strategy. Including the convergence curves of other models in the same panel could provide stronger evidence.

- The proposed WSEL algorithm introduces many weight parameters (e.g., \(\lambda, \gamma, \delta, \beta, \eta, \mu\)), which may require precise fine-tuning to balance the contributions of different components when applied to new datasets. Improper selection of these parameters may lead to degraded model performance.


#### **Minor**:

- To enhance readability, placing the citation next to the name of the method being compared would be better.

- For written rigor, the abbreviations of the six metrics used in Table 3 should be defined in advance, such as Hamming Loss (HL), Ranking Loss (RL), Coverage (CV), Average Precision (AP), Macro-F1 (MA), and Micro-F1 (MI).

**Suitability:**

2

---

### Official Review · Reviewer_KZD6 · 2024-06-03

**Rating:** 5
**Confidence:** 4

**Summary:**

This paper presents a novel EEG feature selection model with weighted self-representational learning (WSEL) for incomplete multi-dimensional emotion recognition. The label space is successfully reconstructed under the incomplete information. Extensive experiments validate the model on two publicly available datasets, and its performance is compared with nine state-of-the-art feature selection approaches.

**Strengths:**

1、An embedded feature selection method for EEG signals is proposed.
2、The consideration of incomplete information in practical applications for EEG analysis is an important aspect for extending affective brain-computer interface (aBCI) techniques into real-world applications.

**Limitations:**

1、The paper template does not meet the requirements of ACM MM 2024.
2、In Fig. 1, there are some typos, such as "Incmoplete" and "Redunancy".
3、An incorporation of a graph-based manifold regularizer is adopted based on the label space. The regularization effect should be thoroughly examined, especially when the emotional labels are limited.
4、The rationale for calculating global feature redundancy learning based on correlation should be explained.
5、Additional discussions on the impact of hyperparameters should be included in the discussion section.

**Suitability:**

3

---

### Meta-Review · Area_Chair_sKbM · 2024-07-01

**Recommendation:** Accept (Oral)
**Confidence:** 4

**Metareview:**

The paper presents a novel EEG feature selection model with weighted self-expression learning (WSEL) aimed at addressing the problem of incomplete multi-dimensional emotional labels in EEG emotion recognition. The model's effectiveness is validated using two datasets, DREAMER and DEAP, and compared against nine state-of-the-art feature selection approaches, showing superior performance in six performance metrics.

Reviewer KZD6's and reviewer xq6x's concerns have been resolved. Reviewer sSZj's concerns have been partially resolved, with some contradictions and a suggestion for improving data presentation. For the final version, please address the contradictions highlighted by Reviewer sSZj regarding the use of subjective labels and improve data presentation as suggested. Additionally, ensure that Reviewer r3sa's concerns are adequately addressed to enhance the overall quality and clarity of the paper.